# Natural Coumarin Derivatives Activating Nrf2 Signaling Pathway as Lead Compounds for the Design and Synthesis of Intestinal Anti-Inflammatory Drugs

**DOI:** 10.3390/ph16040511

**Published:** 2023-03-30

**Authors:** Luiz C. Di Stasi

**Affiliations:** Laboratory of Phytomedicines, Pharmacology and Biotechnology (PhytoPharmaTech), Department of Biophysics and Pharmacology, São Paulo State University (UNESP), Botucatu 18618-689, SP, Brazil; luiz.stasi@unesp.br; Tel.: +55-14-38800216

**Keywords:** IBD, Nrf2, NF-κB, plant-derived coumarins, gut microbial coumarin metabolites, intestinal inflammation

## Abstract

Nrf2 (nuclear factor erythroid 2-related factor 2) is a transcription factor related to stress response and cellular homeostasis that plays a key role in maintaining the redox system. The imbalance of the redox system is a triggering factor for the initiation and progression of non-communicable diseases (NCDs), including Inflammatory Bowel Disease (IBD). Nrf2 and its inhibitor Kelch-like ECH-associated protein 1 (Keap1) are the main regulators of oxidative stress and their activation has been recognized as a promising strategy for the treatment or prevention of several acute and chronic diseases. Moreover, activation of Nrf2/keap signaling pathway promotes inhibition of NF-κB, a transcriptional factor related to pro-inflammatory cytokines expression, synchronically promoting an anti-inflammatory response. Several natural coumarins have been reported as potent antioxidant and intestinal anti-inflammatory compounds, acting by different mechanisms, mainly as a modulator of Nrf2/keap signaling pathway. Based on in vivo and in vitro studies, this review focuses on the natural coumarins obtained from both plant products and fermentative processes of food plants by gut microbiota, which activate Nrf2/keap signaling pathway and produce intestinal anti-inflammatory activity. Although gut metabolites urolithin A and urolithin B as well as other plant-derived coumarins display intestinal anti-inflammatory activity modulating Nrf2 signaling pathway, in vitro and in vivo studies are necessary for better pharmacological characterization and evaluation of their potential as lead compounds. Esculetin, 4-methylesculetin, daphnetin, osthole, and imperatorin are the most promising coumarin derivatives as lead compounds for the design and synthesis of Nrf2 activators with intestinal anti-inflammatory activity. However, further structure–activity relationships studies with coumarin derivatives in experimental models of intestinal inflammation and subsequent clinical trials in health and disease volunteers are essential to determine the efficacy and safety in IBD patients.

## 1. Introduction

Coumarins, also known as benzopyrones, comprise a class of phenolic compounds derived from cinnamic acid and composed of benzene fused to an α-pyrone ring. Natural coumarin and its derivatives are widespread and found in some fungi including Basidiomycetes and Ascomycetes, a lot of plant species classes (edible, medicinal, and spices) belonging to different botanical families, and as metabolites derived from the gut microbiota fermentative process on the different constituents of plant diet commonly used in human feeding. *Dipteryx odorata* Wild. belonging to Fabaceae botanical family is a medicinal plant popularly known in the Brazilian Amazon Forest as “cumaru” (coumarou), which originated the name coumarin. From seeds of *Dipteryx odorata*, named tonka beans, coumarin was first isolated by Vogel in 1820 [1]. Tonka bean oil is rich in coumarin, which has a pleasant smell to resemble vanilla and has been widely used, since 1882, in perfumery and cosmetics as a fixative or to highlight the scent [2,3]. Coumarin is also added to other products including toilet soaps, hair sprays, detergents, tobacco, processed foods, and alcoholic beverages such as vermouth and whiskey [2,3].

Coumarin chemical structure was the basic molecule, after oxidation reaction, for the development of dicoumarol (3,3′-methylene-is(4-hydroxycoumarin) initially used as rodenticides. Chemical changes in dicoumarol molecule generated several compounds with anticoagulants properties, mainly warfarin, the main anticoagulant therapeutic drug used to prevent and treat several diseases, including vein thrombosis, atrial fibrillation, myocardial infarction, and pulmonary embolism [4]. Warfarin can be also synthesized from 4-hydroxycoumarin, a natural simple coumarin, by the Michael condensation reaction with benzal-acetone under basic or acid-catalyzed conditions using either water or piperidine [5]. Initially classified as toxic compounds, coumarin derivatives induce no adverse and toxic effects in response to doses that are more than 100 times the maximum human daily intake, indicating coumarin derivatives exposure is considered safe for humans [2]. Coumarin derivatives at lower doses display a wide range of pharmacological activities and therapeutic applications (Figure 1), which are potentially useful for the design and synthesis of new bioactive compounds [3].

Moreover, coumarins have been recognized as a natural, versatile, privileged, and accessible scaffold for the design and development of bioactive compounds with therapeutic, agrochemical, cosmetic, fragrance, and chemical applications [6,7]. The chemical value of coumarin for the development of new drugs is also correlated to the high potential of its basic structure to combine with other bioactive compounds to produce coumarin hybrids and dimers such as coumarin-chalcone, coumarin-imidazole, coumarin-pyrazole, coumarin-triazole, coumarin-benzotriazole, coumarin-isoxazole, coumarin-dihydroartemisinin, coumarin-hydrazine, coumarin-ergosterol, coumarin-ferrocene, coumarin-pyridine, coumarin-pyrimide, coumarin-benzosulfone, coumarin-imine, and coumarin-uracil with several biological activities [8,9,10,11,12]. Besides hybridization and dimerization as strategies for the development of new drugs, natural coumarin derivatives have been highlighted due to their wide range of pharmacological activities (Figure 1), mainly as an antiprotozoal, vasodilator, inhibitors of several enzymes such as cholinesterase, carbonic anhydrase, monoamine-oxidase, serine protease, cyclooxygenase, and lipoxygenase, antitumor, anti-viral, anti-tuberculosis, antifungal, anti-neurodegenerative intestinal anti-inflammatory, and antioxidant [6,7,8,9,10,11,12,13,14,15,16,17,18,19]. Coumarin derivatives can directly promote intestinal anti-inflammatory activity by different mechanisms or display antioxidant properties by modulating the Nrf2 signaling pathway, indirectly inducing intestinal inflammatory activity through the reduction in oxidative stress (Figure 1).

The antioxidant and anti-inflammatory properties of coumarin derivatives have been widely investigated and reported, mainly for those poly-hydroxylated coumarins recognized as efficient antioxidants in biological systems and useful to prevent and treat several diseases related to oxidative stress [6,15,17,20]. Chemically, coumarin derivatives display antioxidant properties acting by different mechanisms such as inhibiting free radical production by activated oxygen metabolites, changing the structural organization of free radical, producing a local decrease of oxygen concentration, interacting with organic radicals, chelating metal ions, and converting peroxides to stable and inactive products [21]. On the other hand, coumarin derivatives also act as modulators of the endogenous antioxidant system constituted by catalase (CAT), superoxide dismutase (SOD), nicotinamide adenine dinucleotide phosphate (NADPH) oxidase (NOX), heme-oxygenase 1 (HO-1), thioredoxin, and glutathione antioxidant systems formed by glutathione (GSH) and its enzymes glutathione peroxidase (GPX), glutathione reductase (GR), glutathione synthetase (GSS), glutathione S-transferase (GST), and γ-glutamyl-cysteine ligase (γ-GCL) [15]. The endogenous antioxidant system is mainly regulated by nuclear factor erythroid 2 (NEF2)-related factor 2, named Nrf2, which protects cells from oxidative stress induced by reactive oxygen species (ROS), reactive nitrogen species (RNS), and environmental damage as well as coordinates detoxification enzymes related to stress conditions [22]. In addition, Nrf2 activation can prevent inflammatory processes, inhibiting the nuclear factor-κB (NF-ΚB) activation, the major transcriptional factor related to pro-inflammatory cytokines production and release [23].

Oxidative stress is a redox imbalance condition in which excessive levels of reactive oxygen and nitrogen species are related to inadequate availability of endogenous antioxidants, which destroy these harmful products from metabolic processes [24]. Excessive levels of ROS and RNS can damage lipids, proteins, DNA, and other macromolecules, leading to inflammation, oxidative stress, and cell death [19]. The imbalance of the redox system is a key factor for the initiation and progression of several human diseases, including metabolic disorders and non-communicable diseases (NCDs) such as Inflammatory Bowel disease (IBD).

Given these facts, this review underpins the important recent advances in the study of coumarin derivatives as modulators of the Nrf2 signaling pathway and their use as promising lead compounds for the development of new drugs useful to control or treat NCDs with an emphasis on IBD, after briefly introducing Nrf2 signaling pathway to control oxidative stress and inflammatory processes and its particular importance for the control, prevention, and treatment of intestinal inflammatory processes. A Medline search was performed to identify relevant bibliography published between 2013 and 2022 using search combined terms including: “nrf2 coumarin”, “HO-1 coumarin”, “nrf2 IBD”, “nrf2 colitis”, “nfr2 coumarin colitis”, “nrf2 IBD coumarin”, HO IBD”, “HO IBB coumarin”, “HO-1 colitis”, “keap1 coumarin”, and “HO-1 colitis coumarin”.

## 2. The Role of Oxidative Stress in NCDs Focusing on IBD

NCDs are a group of human chronic diseases without a definitive pharmacological cure and long-term duration triggered by multifactorial etiological factors [15,19]. NCDs include several human diseases, mainly diabetes, cardiovascular diseases, obesity, asthma, multiple sclerosis, neurodegenerative diseases such as Parkinson’s and Alzheimer’s diseases, cancer, and IBD. NCDs are the main cause of mortality worldwide, promoting 73.4% of global deaths or 41.1 million people in 2017, draining over USD 60 trillion from the global economy from 2011 to 2030 [25,26]. NCDs are projected to increase to 55 million deaths throughout the world by the year 2030, with important challenges to world health care systems and whose prevention was assigned as a high priority by the World Health Organization [27,28].

NCDs, including IBD, have been recognized as chronic disorders with slow progression only becoming identified after the promotion of persistent cellular damage in several target tissues, which are directly affected by biochemical changes related to redox imbalance and generally linked to inflammatory processes [29,30]. Extrinsic etiological factors including physical inactivity, psychological stress, high energy intake, poor sleep and diet, alcohol and caffeine intake, smoking, intestinal microbiota dysbiosis, exposure to ultraviolet radiation, and lack of vitamin D induce chronic oxidative stress, sustainable cellular damage, and inflammation, which contribute with development and maintenance of NCDs, particularly IBD (Figure 2) [29,31]. Therefore, the use of products or therapeutic approaches to reduce cellular oxidative stress can improve the redox balance and reduce cell damage is a potential and efficient strategy to prevent and control the development of several NCDs, such as IBD [29].

IBD is a chronic intestinal inflammatory process consisting of Crohn’s disease (CD) and ulcerative colitis (UC), which are also part of an immune-mediated inflammatory disease [15]. Although the IBD etiology is unclear, its occurrence and development are triggered by the same NCDs extrinsic risk factors, which when combined with genetic predisposition induce a dysregulated immune response as well as dysfunctional intestinal barrier function related to gut dysbiosis [15,31]. These processes are accelerated and perpetuated by persistent exposure to environmental factors, mainly those related to redox imbalance and tissue oxidative stress either in individuals with or without genetic predisposition (Figure 2) [32,33]. Oxidative stress is a key factor in the pathogenesis of IBD and subsequently a prominent target for synthetic or natural compounds to produce antioxidant effects for IBD control and prevention [15,33,34,35,36,37,38]. There is strong evidence that oxidative stress is increased in NCDs, including IBD; therefore, the use of products and therapeutic approaches to reduce the cellular oxidative stress and cell damage are important and efficient strategies for the prevention, control, and remission of symptoms in IBD patients [29,34,36]. Moreover, intestinal anti-inflammatory drugs commonly used in IBD management, including 5-amino salicylic derivatives such as sulfasalazine and mesalazine, corticosteroids such as prednisolone, and immunosuppressants such as azathioprine also exhibit antioxidant properties by different mechanisms by reducing myeloperoxidase activity, avoiding glutathione depletion induced by intestinal damage, scavenging reactive oxygen species, and modulating mitogen-activated protein kinases (MAPKs) [38,39,40]. An antioxidant represents any compound that delays, prevents, or removes oxidative damage in a target molecule, minimizing the exposure to oxygen molecules. Antioxidant defenses include a lot of endogenous enzymatic and non-enzymatic products such SOD, CAT, glutathione enzymatic family related to the production of GSH such as GR, GPX, and GST, the NOX, and the thioredoxin systems. Redox regulation displays a key role in the control of oxidative stress through several signaling pathways, including MAPKs and transcriptional factors that activate the transcription of multiple genes in response to several stimuli, including the activator protein 1 (AP-1), NF-κB, the peroxisome proliferator-activated receptor gamma (PPAR-γ), and Nrf2. AP-1 transcription factor identifies intracellular oxidative stress and regulates the expression of genes in response to cytokines and growth factors related to stress response, cell growth, and differentiation, particularly associated with cancer progression and control [41]. NF-κB pathway is a transcription factor activated by several stimuli, including reactive oxygen species, which regulate the expression of multiple genes related to the production of pro-inflammatory cytokines, mediating the inflammatory response [42]. PPAR-γ is a key ligand transcription factor up-regulated by Nrf2 particularly related to adipogenesis and metabolic regulation but also reported as a notable enhancer of the antioxidant and anti-inflammatory genes [43,44]. Finally, Nrf2 and its bind with antioxidant response elements trigger the transcription of multiple genes related to antioxidant defense, playing a pivotal role to control the expression of genes that coordinates the maintenance of intracellular redox homeostasis and the regulation of inflammatory processes [45].

## 3. Nrf2 and Its Interaction with NF-κB Signaling Pathways to Control Oxidative Stress and Promote Intestinal Anti-Inflammatory Activity

The maintenance of cellular redox balance to regulate the cellular response to stress and inflammation involves a cooperative interplay between Nrf2 and NF-κB signaling pathways [23,44], which can be highlighted to explain simultaneous antioxidant and anti-inflammatory properties induced by antioxidant compounds, including natural coumarin derivatives.

Nrf2 signaling pathway or Keap1-Nrf2-ARE (Kelch-like ECH-Associated protein 1-nuclear factor erythroid 2-related factor 2-antioxidant response element) system has been reported as the master defense mechanism against oxidative stress, which after activation by different products, is useful to control, prevent, and relieve several symptoms of NCDs, including hypertension and cardiovascular diseases, cancer, diabetes, obesity, neurodegenerative and aging diseases, and IBD [46,47,48,49,50,51,52,53]. The Nrf2 is a complex signaling pathway (Figure 3) imbricated with NF-κB transcription factor to simultaneously modulate oxidative stress and inflammatory processes, increasing the expression of antioxidant and detoxification enzymes, chaperones, growth factors, and transport proteins [46]. Gene transcription is mediated after binding of Nrf2 with antioxidant response elements (ARE), DNA sequences that encode several antioxidant defense enzymes, including CAT, SOD, GST, GPX, GR, HO-1, γ-GCL, and thioredoxin enzymes system (Figure 3) [19,22].

The Nrf2 signaling pathway involves Keap1, which functions as an Nrf2 repressor at the basal level (Figure 3). Under the unstressed condition, Nrf2 recruits two Keap1 molecules, which act as an adaptor protein that allows the binding between Nrf2 and Cul3 (Cullin 3), an E3 ligase required for the ubiquitination of lysine and subsequent proteasome degradation [22,46]. Under oxidative stress conditions promoted by drugs and environmental conditions, endogenous Keap1 and Cul3 are uncoupled with subsequent inhibition of ubiquitination of Nrf2 and proteasome degradation, leading to newly synthesized Nrf2 cytosol accumulation and activation with subsequent Nrf2 translocation into the nucleus [23,46]. Into the nucleus, Nrf2 associates with small Maf proteins (sMaf) and binds to ARE of DNA, promoting gene transcription (Figure 3) [23,46]. As commented, there is an interplay between Nrf2, the master signaling pathway that regulates oxidative stress, and NF-κB, the main transcription factor that mediates inflammatory response via the production of pro-inflammatory cytokines [23]. NF-κB is the main regulator of inflammation, which is activated by two routes, the canonical and non-canonical pathways [15,54]. In the canonical pathway (Figure 3), (RelA/p65)/p50 heterodimers are maintained in the cytoplasm at an inactive state by IκBα, a family of inhibitors of NF-κB [15,54]. The IκB inhibitory complex (IKK) is composed of a regulatory IKKγ subunit and two active subunits IKKα and IKKβ [55]. The activation of this complex occurs by membrane ligands such as bacteria and virus metabolites, growth factors, and cytokines, leading to IκB phosphorylation and rapid proteolytic degradation (Figure 3) [56,57]. The activated heterodimeric NF-κB is translocated into the nucleus, where it interacts with NF-κB response elements (NRE), leading to the transcription of multiple genes related to the inflammatory process, mainly pro-inflammatory cytokines (Figure 3) [15,54,57].

The lack of Nrf2 has been associated with an increment in oxidative stress and pro-inflammatory cytokine production because the NF-κB signaling pathway is activated under oxidative stress conditions via phosphorylation and degradation of IκB [58]. In disease conditions, the regulation of Nrf2 and NF-κB signaling pathways are affected and can be used as two targets for the development of new drugs and therapeutic intervention. The complex crosstalk between Nrf2 and NF-κB signaling pathways is two-ways, in which Nrf2 modulates NF-κB and contrariwise [58]. In this interaction, Nrf2 increases glutathione levels and GSH-dependent enzymes (Figure 3), which reduce oxidative conditions and inhibit NF-κB [58]. This interaction has been related to the action of Keap1 diminishing the phosphorylation of IκB (Figure 3) and autophagic degradation with subsequent negative regulation of NF-κB [59]. Additionally, HO-1 produced by Nrf2 has also been reported as the main Nrf2-mediator of NF-κB inhibition [23]. HO-1 cleaves the porphyrin ring of heme into carbon monoxide, Fe^++^, and biliverdin, which is converted into bilirubin, leading to the inhibition of the Nκ-B signaling pathway (Figure 3) [60]. On the other hand, NF-κB also regulates Nrf2 interaction with ARE by differential mechanisms, particularly by the competition of Nrf2 and p65 proteins for the transcriptional co-activator CREB-binding protein-p300 complex (CBP), which acetylates histones, Nrf2 and p65 proteins (Figure 3) [61]. The p65 subunit of NF-κB represses the Nrf2-ARE at the transcriptional level via the reduction of CBP from Nrf2 competitive interaction or promotion of histone deacetylase 2 [62]. Although it is not completely elucidated, the p65 subunit also interacts with Keap1, increasing the Keap1 nuclear abundance and reducing the Nrf2-ARE signaling pathway by translocation into the nucleus [63]. The interplay between Nrf2 and NF-κB involves complex molecular interactions and mechanisms previously reported [23,58,59,63,64]. Although the co-regulation between these two signaling pathways is not completely elucidated, this interplay can support the effects of compounds on the Nrf2 signaling pathway as modulators of oxidative stress with simultaneous anti-inflammatory activity as evidenced by the reduction in the pro-inflammatory cytokine levels, which is mainly coordinated by NF-κB signaling pathway. Therefore, natural inducers of Nrf2 such as natural coumarin derivatives can display simultaneous antioxidant and anti-inflammatory activities with high pharmacological and therapeutic applications to prevent and control IBD and other NCDs (Figure 4).

## 4. Intestinal Anti-Inflammatory Coumarin Derivatives Targeting Nrf2-Keap1 Signaling Pathway

In this section, the studies with coumarin derivatives from different natural sources, which modulated Nrf2 signaling and showed intestinal anti-inflammatory activity were revised. Scientific evidence based on in vivo and in vitro studies were used to identify promising candidates as lead compounds for design and drug development for the prevention and control of the intestinal inflammatory process in IBD patients. Intestinal anti-inflammatory properties of natural coumarin derivatives were recently revised, some with protective effects through oxidative stress modulation [15]. Moreover, the ability of natural coumarins to modulate the Nrf2 signaling pathway as a central mechanism against oxidative stress has been also previously reported [19]. These studies demonstrated coumarin derivatives modulating the Nrf2 signaling pathway and displaying simultaneous intestinal anti-inflammatory activities, effects potentially useful in the management of intestinal inflammatory processes. Coumarin derivatives modulating the Nrf2 signaling pathway and displaying intestinal anti-inflammatory activity include simple coumarins, linear and angular furanocoumarins from plant origin, and coumarin derivatives produced by the fermentative process performed by gut microbiota on the plant-derived products commonly used in human feeding.

### 4.1. Simple Intestinal Anti-Inflammatory Coumarin Derivatives Targeting Nrf2 Signaling

The simple coumarin derivatives with simultaneous effects on the Nrf2 signaling pathway and intestinal inflammation include esculetin, 4-methylesculetin, esculin, daphnetin, umbelliferone, osthole, fraxetin, scopoletin, and scoparone. The previous in vitro and in vivo studies with these simple coumarin were analyzed to identify the most promising natural coumarins as lead compounds for the development of new drugs to control IBD.

#### 4.1.1. Esculetin, 4-Methylesculetin, and Esculin

Esculetin (6,7-dihydroxycoumarin) is found in several plants, mainly in *Fraxinus rhynchophylla* Hance (ash tree) and other *Fraxinus* species belonging to Oleaceae family, whereas its methylated derivative at C4, 4-methylesculetin (6,7-dihydroxy-4-methyl-coumarin), is obtained from several plants and chemical synthesis. Both derivatives are simple antioxidant coumarin derivatives (Figure 5) that modulate the Nrf2 signaling pathway and display several pharmacological activities [65,66]. Esculetin glycosylated at C6, named esculin and known as 6-glucoside-7-hydroxycoumarin (Figure 5) also activates the Nrf2 signaling pathway [67] and is mainly found in *Aesculus hippocastanum* L. (horse-chestnut) medicinal plant belonging Hippocastanaceae family and other plant species.

Esculetin at concentrations of 100, 200, 300, and 500 µM was evaluated on the PANC-1, AsPc-1, and MIA PaCa-2 carcinoma cells and when used at high concentrations inhibited cell growth [65]. This effect was related to a high decrease in ROS generation and the protein levels of NF-κB with a simultaneous increase of Nrf2 nuclear accumulation and upregulation of Nrf2-induced NADPH quinone dehydrogenase 1 (NQO1) expression [65]. The effects of esculetin on the Nrf2 signaling pathway were dependent on both direct binding between esculetin and Keap1 as evidenced by in silico analyses, docking studies, and pull-down assay as well as by high Keap1 phosphorylation associated with activation of ARE interaction with Nrf2 [65]. In the same study, esculetin simultaneously decreased the p65 subunit of NF-κB [65]. Esculetin at concentrations of 12.5, 25, 50, and 100 µM also inhibited the oxidative stress, reducing nitric oxide synthase and nitric oxide levels as well as exhibited an anti-inflammatory response through the reduction in TNF-α and chemoattractant protein-1 (MCP-1) production [66]. These protective effects were related to an induction of HO-1 in co-cultured macrophages (RAW264.7 cells) and 3T3-L1 adipocytes [66]. The dual effects of esculetin activating Nrf2 and inhibiting NF-κB signaling pathways improved the antioxidant imbalance in NB4 leukemia cells at concentrations ranging from 20 to 500 µM [67]. These effects were dependent on the reduction of ROS levels and 5-LOX (5-lipoxygenase) activity with simultaneous increased SOD activity and c-jun NH_2_ terminal kinase (JNK) and p38 MAPKs phosphorylation [67]. Pretreatment of human corneal cells with esculetin at concentrations of 20, 40, 80, and 100 µM promoted antioxidant defense through induction of Nrf2 translocation into the nucleus, upregulating HO-1, NQO1, and SOD gene expression [68]. The counteraction of oxidative stress by esculetin through Nrf2 activation and differential mechanisms of action was also demonstrated in in vitro assays using human neuronal SH-SY5Y cells and C2C12 myoblasts at concentrations ranging from 1.25 to 20 µM [69,70].

Esculetin, when evaluated by different in vivo studies, also produced a series of pharmacological activities related to the control of oxidative stress. The protective effects of esculetin orally administered at a dose of 20 or 40 mg/Kg in experimental lupus nephritis model in MRL/Ipr mice was dependent on simultaneous Nrf2 activation and NF-κB inhibition with subsequent reduction of oxidative stress and pro-inflammatory cytokines production [71]. NF-κB inhibition and Nrf2 activation signaling pathways were accompanied by increased GSH levels and GPX activity after oral administration of 50 mg/Kg esculetin in a model of aluminum chloride-induced male reproductive toxicity in rats [72]. Cognitive impairments in male ICR mice were improved after administration of esculetin at 20 and 80 mg/Kg, which acted upregulating Nrf2 signaling with additional effects on the regulation of mitochondrial fragmentation and mitophagy markers [73]. The pharmacological effects of esculetin to promote intestinal anti-inflammatory activity via the Nrf2 signaling pathway are illustrated in Figure 5.

The activation of the Nrf2 signaling pathway was also reported as a key action for 4-methylesculetin to induce a reduction of body weight, visceral obesity, blood glucose, adipocyte size, and hepatic lipid accumulation, after oral treatment with 15 and 50/Kg by 8 weeks in obese mice by a high-fat diet [74]. Similar to other simple coumarins, the anti-inflammatory effects of esculin were closely related to the Nrf2 activation and subsequent upregulation of HO-1 and NQO1 expression [75]. These effects were mechanistically related to the suppression of Nrf2 ubiquitination and reduction in Nrf2 degradation in RAW264.7 cells [75]. These effects were improved when an esculin transglycosylated derivative was obtained using cellobiose and β-glucosidase, indicating additional glycosylation increases the esculin action on the Nrf2 signaling pathway [75]. Moreover, it was demonstrated in zebrafish cultures and molecular docking studies that esculin inhibits the binding of Keap1 with Nrf2, significantly increasing the Nrf2 target genes, including HO-1 [76]. In an experimental model of lipopolysaccharide/D-galactosamine-induced acute liver damage in BALBc mice, esculin intraperitoneally administered at 10, 20, and 40 mg/Kg has a hepatoprotective effect, reducing MPO activity, MDA content, TNF-α, and IL-1β production through inhibition of NF-κB and upregulation of Nrf2 and HO-1 expression [77].

Esculetin, 4-methylesculetin, and esculin were reported as active intestinal anti-inflammatory coumarin derivatives in the trinitrobenzene sulphonic acid (TNBS) or dextran sulfate sodium (DSS) experimental models, protecting intestinal damage through reduction of the oxidative stress as evidenced by a reduction in myeloperoxidase (MPO) activity and counteraction of the GSH depletion induced by intestinal damage [78,79,80]. These effects were observed in the TNBS model of intestinal inflammation in rats after oral administration of 10 mg/Kg of esculetin and 2.5, 5, and 10 mg/Kg of 4-methylesculetin, which were more pronounced than those effects promoted by 25 mg/Kg of sulphasalazine, a reference drug to treat UC in human [78]. Intestinal inflammation induced by TNBS in male Sprague Dawley rats also was alleviated after intra-rectal administration of 100 and 200 µM of esculetin, which reduced intestinal damage, MPO, cyclo-oxygenase 2 (COX-2), and inducible nitric oxide synthase (iNOS) activities with no differences between doses [81]. When evaluated in human colon carcinoma (HCT116), human embryonic kidney 293, and human renal cancer UMNRC2 cells, esculetin at concentrations of 25, 50, and 100 µM activated hypoxia-inducible factor-1α (HIF-1α) via inhibition of HIF prolyl hydroxylase-2 (HPH-2), an enzyme responsible by negative regulation of HIF-α stability [81]. The intestinal anti-inflammatory activity of esculetin (25 mg/Kg by oral route) in mice DSS-induced intestinal inflammation model was related to a reduction in MPO activity, counteraction of GSH depletion, and reduction in IL-6 production [80]. In the same experimental model of intestinal inflammation using C57BL/6 female mice, esculetin orally administered at 20 mg/Kg ameliorated intestinal injury, decreased MPO activity and IL-6 and TNF-α production, and inhibited NF-κB/MPAKs signaling pathways [82]. The inhibitory action of the NF-κB activation, p38, JNK, and extracellular signal-regulated kinase (ERK) phosphorylation, and IL-6, nitric oxide (NO), and TNF-α production were corroborated in RAW264.7 cells treated with 10, 25 and 50 µM of esculetin [82].

Moreover, 4-methylesculetin at doses of 5 and 10 mg/kg by oral route upregulated the GST and GR as well as prevented the Nrf2 downregulation induced by oxidative stress and intestinal inflammatory process induced by TNBS in rats [83]. 4-methylesculetin interacted at molecular levels with glutathione reductase, stabilizing its enzymatic activity and reducing oxidative stress [83]. Although 4-methylesculetin was not reported as an inhibitor of the NF-κB signaling pathway, a reduction of pro-inflammatory cytokines production, such as IL-1β, IL-6, IL-17, and TNF-α was described in experimental models of intestinal inflammation at same doses [79,84]. Oral administration of esculin at doses of 5, 10, and 25 mg/Kg with TNBS-induced intestinal inflammation in rats alleviated the symptoms of gut damage, reducing MPO activity and counteracting GSH depletion when treated with 25 mg/Kg [84]. Intestinal anti-inflammatory activity of esculin (5 mg/kg by intraperitoneal route) was also reported in the DSS model of intestinal inflammation in BALBc mice, protecting intestinal damage, inhibiting NF-κB signaling with reduced production of pro-inflammatory cytokines and simultaneous increase in nuclear localization of PPAR-γ [85]. The pharmacological effects of 4-methylesculetin and esculin to promote intestinal anti-inflammatory activity via the Nrf2 signaling pathway are illustrated in Figure 5.

#### 4.1.2. Daphnetin

Daphnetin (7,8-dihydroxy-coumarin) is found in several plant species, particularly as the main coumarin derivative in *Daphne odora* Thumb. (winter daphne) belonging to the Thymelaeaceae family. Daphnetin is a simple coumarin (Figure 6) with a wide range of pharmacological activities, acting by different mechanisms mainly as an antioxidant agent via modulation of the Nrf2 signaling pathway. Daphnetin at doses of 20, 40, and 80 mg/kg by intraperitoneal route, protected animals against lipid peroxidation, improving enzymatic (SOD, CAT, GPX) and GSH antioxidants markers in a model of 7,12-dimethylbenz(a)anthracene-induced mammary carcinogenesis in female Sprague Dawley rats [86]. These anti-oxidative properties were related to the upregulation of Keap1-Nrf2 associated with HO-1 expression with synchronized downregulation of protein kinase B (Akt)-mediated NF-κB expression [86]. Similar results were reported in a model of cisplatin-induced nephrotoxicity in C57BL/6 mice, in which daphnetin at doses of 2.5, 5, and 10 mg/kg by intraperitoneal route upregulated Nrf2 and HO-1 expressions with simultaneous downregulation of NF-κB and subsequently the reduction in TNF-α and IL-1β pro-inflammatory cytokines production [87]. Recently, the interplay between Nrf2 and NF-κB signaling pathways was also reported as the main mechanism of daphnetin action to inhibit spinal glial activation and to attenuate inflammatory pain, after administration of 4 and 8 mg/kg (intraperitoneal route) in ICR mice [88]. These effects were associated with a downregulation of expression and reduction of levels of IL-6, IL-1β, and TNF-α [88]. The control of allergic rhinitis induced by ovalbumin in C57BL/6 female mice after oral treatment with 5 mg/Kg alleviated nasal symptoms, inflammatory response, and oxidative stress, activating Nrf2/HO-1 and inactivating the NF-κB pathway through the reduction in Toll-like Receptor 4, NF-κB protein levels, and TNF-α and IL-5 production [89].

Modulation of Nrf2 associated with other signaling pathways has been reported as the molecular mechanism of protection against several diseases, including cardiac hypertrophy and fibrosis [90], diabetes, lipid metabolism, and insulin resistance [91,92], hepatotoxicity [93], inflammation [88], lung toxicity [94], and neuronal and renal damage [95,96,97]. Protective antioxidant properties of daphnetin upregulating the Nrf2/HO-1 pathway with simultaneous inhibition of transforming growth factor β1 (TGF-β1)/Smad2/3 signaling axis was demonstrated in rat cardiomyoblast H9c2 cells and transverse aortic constriction model in C57BL/6 mice, alleviating cardiac hypertrophy and fibrosis [90]. Hepatoprotective effects were reported on HepG2 cells and acute liver failure experimental models using C57BL/6 mice, in which daphnetin enhanced the Keap1/Nrf2 signaling, upregulating thioredoxin-1 expression, inactivating thioredoxin-interaction protein, and suppressing apoptosis signaling-regulating kinase/JNK pathways [93]. Daphnetin at 40 mg/Kg also reduced renal toxicity in a model of gentamicin-induced renal injury in ICR mice upregulating the Nrf2 and NOX-4 and reducing Bcl-2-like protein 4 and caspase-3 [96]. Several in vitro studies corroborated the effects of daphnetin activating the Nrf2 signaling pathway and its action on other mechanisms, including the stabilization of anti-apoptotic factor B-cell lymphoma-2 with simultaneous activation of adenosine-5′-monophosphate (AMP)-activated protein kinase, JNK, and ERK phosphorylation in human lung epithelial cells [94]; downregulation of the sterol regulatory element-binding protein-1C, patatin-like phospholipase domain-containing protein, cytochrome p450 2E1, and cytochrome P450 4A; upregulation of phosphoinositide 3-kinase (PI3K) and AMP kinase phosphorylation and increase of Akt levels to control lipid metabolism, insulin resistance, and oxidative stress in an in vitro model of non-alcoholic fatty liver disease [92]. Moreover, daphnetin inhibited glucose-induced extracellular matrix components, oxidative stress, and inflammation in human glomerular cells through Nrf2 activation and simultaneous inhibition of Akt and p65 levels [91]. Daphnetin also modulated the Nrf2 signaling pathway activating JNK and ERK with a subsequent increment of Nrf2 nuclear translocation [98]. This mechanism was related to the effects of daphnetin at concentrations of 2.5, 5, and 10 µM reducing ROS generation and malondialdehyde (MDA) formation and increasing SOD levels and GSH/GSSG (oxidized GSH) ratio [98]. These effects were associated with the inhibition of tert-butyl hydroperoxide-stimulated oxidative damage, cytotoxicity, cell apoptosis, and mitochondrial dysfunction in RAW264.7 cells and peritoneal macrophage from WT and Nrf2−/− mice [98]. Daphnetin mechanisms of action to promote neuroprotective effects acting on Nrf2 and other signaling pathways were also recently revised [99].

As demonstrated, daphnetin displays several mechanisms of action associated with the modulation of the Nrf2 signaling pathway and inhibition of NF-κB signaling, reducing pro-inflammatory cytokines transcription (Figure 6). Together, the studies showed daphnetin as a very important coumarin derivative displaying a wide range of pharmacological activities related to several diseases where oxidative stress takes place as a key etiologic factor, including NCDs. Daphnetin was reported as a potent intestinal anti-inflammatory coumarin derivative in the TNBS model of intestinal inflammation in rats, reducing gut damage lesions, MPO, and alkaline phosphatase activities, and counteracting GSH depletion when orally administered at doses of 2.5, 5, and 10 mg/Kg [84]. In the DSS model of intestinal inflammation using BALB/c mice, daphnetin at 4, 8, and 16 mg/Kg orally administered, ameliorated gut macroscopic and microscopic damages, downregulated TNF-α, CXCL1, and CXCL2, upregulated IL-10 anti-inflammatory cytokine, and reversed DSS-induced gut dysbiosis, increasing the short-chain fatty acid-producing bacteria [100]. In the same study, it was demonstrated daphnetin regulates colonic immune responses and intestinal integrity through the upregulation of zonulin-1, occluding, mucin 2, and E-cadherin and downregulation of IL-1β, IL-6, IL-21, and IL-23 inflammatory cytokines [100].

#### 4.1.3. Osthole

Osthole (7-methoxy-8-isopentenoxycoumarin) is a simple coumarin derivative (Figure 7) originally isolated from fruits of *Cnidium monnieri* (L.) Cusson ex Juss. (Cnidii Fructus or Monnier’s snow-parsley) and also found in other plant species belonging to the Apiaceae family such as *Angelica archangelica* L. and *Angelica pubescens* Maxim. and Rutaceae family such as *Murraya alata* Drake, *Pentaceras australe* (F. Muell.) Benth. Osthole displays a lot of pharmacological properties, mainly acting as a neuroprotective, anti-inflammatory, anticancer, antioxidant, and hepatoprotective agent [101,102]. Intestinal anti-inflammatory and antioxidant properties of osthole were evidenced by in vitro and in vivo assays suggesting its potential use to control and prevent IBD [15]. Several pharmacological activities of osthole have been closely related to the regulation of the Nrf2 signaling pathway and subsequent enhancement of endogenous antioxidant products such as HO-1 and NQO1 with simultaneous inhibition of NF-κB pathway to reduce pro-inflammatory cytokines production [103,104].

Osthole (25, 50, and 100 mg/Kg by intraperitoneal route) administered in C57BL/6 mice submitted to transient global cerebral ischemia/reperfusion model of cognitive impairments, improved the cognitive ability and enhanced pyramidal neurons in hippocampal region with simultaneous increase in SOD activity and decrease in MDA level [105]. Using HT22 murine hippocampal neuronal cells, the authors also demonstrated osthole (25, 50, and 100 µM) increasing Nrf2 and HO-1 protein levels in a concentration-dependent response [105]. The protective effects of osthole on glutamate-induced apoptosis in HT22 cells and Alzheimer’s disease model in APP/PSI mice were also related to the modulation of oxidative stress [106]. On the HT22 cells, osthole at concentrations of 20 and 40 µM protected glutamate-induced cell damage, reducing caspase-3, caspase-8, and caspase-9 activities and upregulating Nrf2, HO-1, and SOD expression, which was related to a reduction in Keap1, effects corroborated after administration of 15 and 30 mg/Kg in Alzheimer’s disease model in mice [106]. Similar results were reported after osthole administration at a dose of 25 mg/Kg in chronic sleep deprivation-induced memory deficits in rats [107]. Osthole at doses of 30 and 40 mg/Kg in rats previously feed high-fat/high-sugar diet improved metabolic syndrome and reduced IL-1β and IL-6 levels, improving kidney dysfunction, oxidative stress, and lipid accumulation via stimulation of Nrf2 expression with subsequent upregulation of SOD, CAT, and GPX production [108]. Nephroprotection was also reported after administration of 30 mg/Kg osthole in female B cell deficiency mice, improving renal function and reducing renal progressive lesions, effects associated with a reduction of ROS generation and increase of Nrf2 nuclear translocation with synchronized inhibition of NF-κB activation [109]. The enhancement of the Nrf2 antioxidant signaling and inhibition of NF-κB was also described in BALB/c mice model of accelerated focal segmental glomerulosclerosis, in which, osthole at a dose of 30 mg/Kg (intraperitoneal route) improved renal function, reducing ROS generation, increasing HO-1 protein levels and GPX activity, and inhibiting the expression of cyclooxygenase-2 and prostaglandin production [104].

Anti-inflammatory activity of osthole in several in vitro and in vivo studies has been reported, suggesting differential mechanisms of action by different signaling pathways. In vitro studies using LPS-induced BV2 mouse microglial cells were incubated with different concentrations of osthole (4, 7, and 10 µg/mL), inducing a reduction in IL-6, TNF-α, and IL-1β levels via inhibitory action of NF-κB pathway and an increase of HO-1 protein via Nrf2 activation [110]. Inflammatory response on the stimulated HepG2 and 3T3-L1 preadipocyte cells was also attenuated by osthole at concentrations of 3, 10, 30, and 100 µM, which inhibited IL-1β, IL-6, and IL-8 levels and expression and reduced COX-2 protein expression [111]. These findings were related to a decrease of MAPKs ERK, p38, JNK, and IκBα phosphorylation with increased HO-1 protein levels [111]. LPS-induced Caco2 human colorectal adenocarcinoma cells and Caco-2/THP-1 and Caco2/macrophages co-cultures were evaluated after incubation with 150, 300, and 450 ng/mL of osthole, which reduced the secretion of IL-1β, IL-6, IL-8, and TNF-α proinflammatory cytokines through inhibition of LPS-induced NF-κB activation and COX2 expression [112]. Osthole at 100 mg/kg by intraperitoneal route attenuated several clinical and histopathological indicators in the TNBS model of intestinal inflammation in C57BL/6 mice, significantly reducing the IL-1β, TNF-α, IL-6, and COX-2 gene expression and improving intestinal barrier function via upregulation of claudin-1 and zonulin-1 genes [113]. Using the model experimental of intestinal inflammation induced by dinitrobenzene sulphonic acid (DNBS) in rats, oral administration of 50 mg/Kg of osthole reduced TNF-α and increased anti-inflammatory IL-10 cytokine, with no effects on the INF-γ levels [114]. Osthole also displayed an antioxidant property, reducing MDA levels and MPO activity, increasing GPX, CAT, SOD, and GST levels, and counteracting the GSH depletion induced by TNBS [114]. Moreover, in the DSS model of intestinal inflammation in BALBc mice, osthole showed protective effects on intestinal inflammation improving clinical parameters and histological damages and reducing MPO activity and colon TNF-α expression [115]. These findings were corroborated using RAW164.7 cell cultures, in which osthole at concentrations of 12.5, 25, 50, or 100 µM promoted several actions, inhibiting LPS-induced NO, COX-2, PGE_2_, TNF-α, and IL-6 [115]. The intestinal anti-inflammatory activity of osthole was closely related to a significant reduction in ERK and p-38 MAPKs and an increase in IκBα degradation [115]. The attenuation of p38 phosphorylation was previously reported in the TNBS model of intestinal inflammation and RAW264.7 cells [113]. The pharmacological effects of osthole to promote intestinal anti-inflammatory activity via the Nrf2 signaling pathway are illustrated in Figure 7.

#### 4.1.4. Umbelliferone

Umbelliferone (Figure 8) is a simple coumarin with widespread occurrence in several botanical families such as Apiaceae, Asteraceae, Acanthaceae, and Hydrangenaceae. Additionally, known as 7-hydroxycoumarin, hydrangine, and skimmetine, this coumarin derivative displays several pharmacological activities. Among its actions, umbelliferone attenuates oxidative stress modulating the Nrf2 signaling pathway to control several disorders, mainly hepatotoxicity, diabetes, inflammation, allergy, and renal and cardiovascular disorders [116,117,118,119,120].

In vivo studies demonstrated umbelliferone protecting rats against tetrachloride-, cyclophosphamide-, methotrexate-, and N-nitrosodiethylamine-induced hepatotoxicity in rats, alleviating hepatic damage in db/db mice and hepatic ischemia/reperfusion-induced oxidative stress in rats via activation and upregulation of Nrf2 signaling pathway [116,117,120,121,122,123]. Hepatoprotective effects of umbelliferone at doses of 30, 40, or 50 mg/Kg were related to the modulation of Nrf2 signaling with subsequent upregulation of HO-1, NQO1, GLC, CAT, GPX, and SOD genes and GSH content and reduction in NO and ROS generation [116,121,122]. Umbelliferone also displays protection against metrothexate- and cisplatin-induced nephrotoxicity in mice and rats, reducing oxidative stress by restoration of GSH levels, SOD, and GST activities and reduction of ROS generation via Nrf2 signaling [124,125,126]. Similar to other coumarin derivatives, umbelliferone also inhibits the NF-κB signaling pathway, reducing the production or expression of pro-inflammatory cytokines, such as IL-1β, TNF-α, and IL-6 and increasing IL-10 anti-inflammatory cytokine associated with MAPK and PPAR-γ signaling pathways [117,121,124]. Moreover, umbelliferone at 15 mg/kg by intraperitoneal route, attenuated streptozotocin-induced cognitive dysfunction in rats, reducing oxidative stress and neuroinflammation through activation of the Nrfr2/HO-1 signaling pathway [127]. Oral administration of 30 mg/Kg of umbelliferone in rats with acetic acid-induced intestinal inflammation improved the macroscopic and microscopic damage, reduced colon TNF-α, IL-6, MPO, and downregulated TLR4, NF-κB, and iNOS inflammatory factors, via upregulation of PPARγ and sirtuin 1 signaling pathways [128].

#### 4.1.5. Fraxetin

Fraxetin (7,8-dihydroxy-6-methoxycoumarin), also isolated from seeds of *Fraxinus rhynchophylla* Hance (ash tree)*,* is a simple coumarin (Figure 8) with anti-inflammatory and antioxidant properties, exerting some pharmacological actions by modulating the Nrf2 signaling [129,130]. Antioxidant properties of fraxetin at 10, 25, 50, and 100 µM was reported on HaCaT cells, upregulating HO-1 gene expression by AKT or AMPK pathways and directly associated with an increase in Nrf2 levels via localization of Nrf2 into the nucleus and increasing ARE gene activity [129]. In BALB/c mice, 5 and 25 mg/Kg of fraxetin reduced lipid peroxidation and increased GSH levels and CAT, SOD, GST, GR, and GPX enzymatic activity through positive modulation of Nrf2 levels [130]. Fraxetin inhibited oxidative stress and inflammatory markers such as TNF-α and IL-1β via upregulation of HO-1 protein in ethanol-induced hepatic fibrosis in rats [131]. Although fraxetin-8-O-glucoside (fraxin) modulated Nrf2 pathway-dependent HO-1 expression and promoted anti-inflammatory activity, its effects on intestinal inflammatory processes have not been reported [19]. On the other hand, the intestinal anti-inflammatory activity of fraxetin (5, 10, and 25 mg/Kg, oral route) was reported in the TNBS model of intestinal inflammation in rats, ameliorating intestinal damage, reducing lipid peroxidation, and counteracting the GSH depletion induced by intestinal damage [84].

#### 4.1.6. Scopoletin and Scoparone

Other simple coumarin derivatives with antioxidant and intestinal anti-inflammatory properties are scopoletin and scoparone (Figure 8). Scopoletin (6-methoxy-7-hydroxycoumarin) is mainly found in the roots of several plants, mainly *Scopolia carniolica* Jacq. (henbane bell) and *Scopolia japonica* Maxim. (Japanese belladonna) belonging to the Solanaceae family, whereas scoparone (6,7-dymethoxycoumarin) is obtained from *Artemisia scoparia* Waldest & Kit. (virgate wormwood) belonging to the Asteraceae family and two Rutaceae plant species *Haplophyllum ramosissimum* (Paulsen) Vved. and *Haplophyllum thesioides* (Fisch ex DC.) G.Don. The pharmacological effects of scopoletin and scoparone have been related to the positive regulation of the Nrf2 signaling pathway. Scopoletin at 10 and 30 mg/Kg (oral route) protected rats against methylglyoxal-induced hyperglycemia and insulin resistance via Akt phosphorylation and upregulation of Nrf2 and PPARγ signaling pathways [132]. The activation of Nrf2 with simultaneous inhibition of NF-κB signaling pathways was described as the main mechanism of scopoletin when administered at a dose of 50 mg/Kg by intraperitoneal route) to induce protection against vancomycin-induced intoxication in rats [133]. Scoparone (20, 40, and 80 mg/Kg) also regulated the ROS generation via Nrf2 activation to improve hepatic inflammation in an in vivo model of nonalcoholic fatty liver disease-nonalcoholic steatohepatitis in mice and LPS-induced rAW264.7 cells [134]. Both simple coumarin derivatives, after oral administration (5, 10, and 25 mg/Kg), promoted intestinal anti-inflammatory activity in the TNBS model of intestinal inflammation in rats, counteracting GSH depletion with no effects on MPO activity [84]. Moreover, scopoletin was evaluated in the TNBS model of intestinal inflammation in Sprague Dawley rats, but the intestinal anti-inflammatory effects were not related to the inhibition of the HPH-2 enzyme and HIF-1α pathway [81].

### 4.2. Intestinal Anti-Inflammatory Furanocoumarin Derivatives Targeting Nrf2 Signaling

Imperatorim also known as ammidin, marmelosin, or marmelide, is a linear furanocoumarin isolated from several plants belonging Apiaceae family such as *Angelica archangelica* L., *Angelica dahurica* Fisch. Ex Hoffm.m and *Glehnia littoralis* F. Schimidt ex Miq. and other plant species from different genera and botanical families. Imperatorin (8-isopentenyloxypsoralen) is a psoralen derivative containing an oxy-isopentenyl group at C-8 (Figure 8). Psoralen, the basic structure of linear furanocoumarins, is found in *Psoralea corylifolia* L. (Babchi) and several food plants, including *Apium graveolens* L. (celery), *Foeniculum vulgare* Mill. (fennel), *Daucus carota* L. (carrot) belongs to the Apiaceae family, and several Rutaceae plant species, such as *Ficus carica* L. (Figure 8). Psoralen and imperatorin have been reported as active natural coumarin derivatives able to modulate the Nrf2 signaling pathway promoting beneficial effects useful to prevent and control several NCDs, including IBD [15,19,135,136,137].

Psoralen administered by gavage at a dose of 20 mg/Kg upregulated the Nrf2 signaling pathway protecting mice against radiation-induced bone injury [135]. Psoralen also ameliorated DSS-induced intestinal inflammation in C57BL/6 mice after intraperitoneal administration of 3 mg/Kg, inhibiting NLRP3 inflammasome, caspase-1, and IL-1β gene expression [137]. However, psoralen induces hepatotoxicity at a dose range of 20 to 800 mg/Kg in mice [138,139,140], limiting its use as a drug.

On the other hand, imperatorin effects on the Nrf2 have been also reported in several in vitro and in vivo studies. Treatment with imperatorin at concentrations ranging from 10 to 100 µg/mL in arsenic trioxide-induced cytotoxicity in H9c2 cells suppressed ROS generation and increased Nrf2, NQO1, and HO-1 expression and protein levels [141]. Imperatorin also suppressed allergic response in peritoneal rat mast cells through Nrf2/HO-1 activation and MAPK and NF-κB inhibition [142]. The upregulation of Nrf2 with reduction of oxidative stress and the inflammatory process was reported after oral administration of imperatorin at 15 and 30 mg/Kg in Sprague Dawley rats submitted to high-fat/high-fructose diet-induced cardiac remodeling and dysfunction [143]. In BALB/c mice asthma model using ovalbumin, imperatorin at 15, 30, and 60 mg/Kg regulated several signaling pathways, increasing the nuclear Nrf2 and HO-1 levels with a simultaneous reduction in cytosol Nrf2 and NF-κB, AKT, Erk, p-38 and JNK levels [144].

Several studies using TNBS and DSS experimental models of intestinal inflammation demonstrated the intestinal anti-inflammatory properties of imperatorin, suggesting its potential to control and prevent IBD. In the DSS model in mice, 25, 50, and 100/Kg of imperatorin attenuated macroscopic and microscopic intestinal damage, avoiding body weight loss and bloody diarrhea and reducing macroscopic and microscopic scores of the lesion [145]. The effects of imperatorin at concentrations of 6.25, 12.5, and 25 µM were also evaluated on the human intestinal epithelial HCT116, LS174T, human leukemia THP-1, and HEK293T cell lines as an attempt to elucidate the main actions of imperatorin, which produced a range of effects, mainly acting as an agonist of pregnane X receptor and inhibiting the NF-κB-mediated TNF-α, IL-1β, and IL-6 pro-inflammatory cytokines production [145]. The intestinal anti-inflammatory activity of imperatorin was also demonstrated in the TNBS model and associated with several pharmacological mechanisms. Imperatorin administered by an intraperitoneal route at concentrations of 15, 30, and 60 mg/Kg in Sprague Dawley rats ameliorated the macroscopic and microscopic intestinal damage induced by TNBS as well as reduced the colon levels of TNF-α and IL-6, upregulating the Nrf2, ARE, and HO-1 expression [136]. Moreover, imperatorin was reported as the main active component of *Angelica dahurica* and *Angelica albicans* plant extracts, which were able to reduce TNBS-induced intestinal inflammation [146,147].

### 4.3. Intestinal Anti-Inflammatory Gut Microbial Coumarins Targeting Nrf2 Signaling

A remarkable group of natural coumarins derived from gut microbiota enzymatic action and named urolithins has been highlighted in recent years due to their pharmacological actions, particularly antioxidant properties. Urolithins are benzocoumarins derived from diphenylpyran-6-one and classified as a combination of coumarin and isocoumarin chemical structure [148,149]. Urolithins include several penta-, tetra-, tri-, di-, and monohydroxylated compounds dependent on the level of hydroxylation on the ellagitannins from the diet promoted by gut bacteria, mainly Gordonibacter urolithinfaciens and Gordobacter pamelaceae, on the ellagitannins from the diet [149]. Other gut bacteria that can participate in the production of urolithins are the *Ellagibacteris isourilithinifaciens* and strains of *Bifidobacteria*, mainly *Bifidobacterium pseudocatenulatum* [150]. The main dietary sources of ellagitannins include pomegranate (*Punica granatum* L., Lythraceae botanical family) fruits, several nuts, mainly walnuts (*Juglans regia* L., Junglandaceae botanical family), and several species of raspberries, mainly red raspberries (*Rubus idaeus* L., Rosaceae family) and black raspberries (*Rubus occidentalis* L.), and black tea (*Camelia sinensis* (L.) Kuntze, Theaceae botanical family). Ellagitannins are firstly converted to ellagic acid by tannases for further production of intermediate luteic acid, which generated urolithin M5, the key precursor of several bioactive urolithins, mainly the bioactive urolithins A and urolithin B (Figure 9) [150,151]. Urolithin A and urolithin B display several pharmacological properties, including anti-cancer, neuroprotective, hepatoprotective, nephroprotective, anti-metabolic, anti-inflammatory, and against autoimmune, cardiovascular, genetic, and aging-associated diseases acting by different signaling pathway modulation [148,150].

Urolithin A and urolithin B modulate oxidative stress and are recognized as emerging antioxidant coumarin derivatives, reducing ROS generation, free radical scavenging, and activating the Nrf2 signaling pathway through Nrf2 nuclear translocation with further upregulation of HO-1, SOD, glutathione-related antioxidant system, and NQO1 [148]. The antioxidant properties of urolithin A and urolithin B were related to the control of several diseases such as diabetes, skin aging, sclerosis, kidney and liver toxicity, inflammation, and osteoclastogenesis [152,153,154,155,156,157,158,159].

Urolithin A at a concentration of 10 µM on high glucose-induced human retinal endothelial (HRE) cells counteracted the oxidative stress, increasing SOD activity and GSH levels and reducing MDA, IL-6, IL-1β, and TNF-α levels and gene expression [152]. The antioxidant properties were related to the activation of Nrf2 and HO-1 levels and Nrf2 activity in HRE cells as well as in streptozotocin-induced diabetic Sprague Dawley rats treated with intraperitoneal administration of 2.5 mg/Kg/day for 12 weeks with urolithin A [152]. In human dermal fibroblasts exposed to ultraviolet A (UVA) radiation, pretreatment with urolithin A at a concentration of 0.2 µM ameliorated UVA-induced proliferative fibroblast dysfunction, protected fibroblast from DNA damage, promoted ROs scavenging activity and Nrf2 activation subsequently driving the activation of antioxidant enzymes, corroborating the antiaging properties of urolithin A previously reported on senescent human skin fibroblasts [152,160]. In vivo studies also demonstrated urolithin A producing several pharmacological activities via Nrf2 activation. The anti-atherosclerotic activity was reported after oral administration of urolithin A at a dose of 3mg/Kg/day for three weeks in rats with a diet rich in cholesterol and subjected to balloon injury of the aorta [156]. Oral administration of urolithin A (20, 50, and 100 mg/Kg/day for 7 days) in C57BL/6 mice subjected to renal/ischemia surgery model alleviated kidney injury via the Keap1-Nrf2 pathway [155]. The hepatoprotective effect was also reported in an acetaminophen hepatotoxicity model in mice, in which urolithin A at intraperitoneal doses of 50, 100, and 150 mg/Kg inhibited oxidative stress accumulation and activated the Nrf2 pathway [154]. Moreover, urolithin A (25 mg/Kg, oral administration) in a model of LPS-induced osteoporosis in C57BL/6 mice attenuated osteoclastogenesis through simultaneous regulating of p28 MAPK and Nrf2 signaling pathways [157].

Urolithin B displayed cardioprotective effects and anti-inflammatory and anticlastogenesis activities modulating the Nrf2 signaling pathway as described by in vitro and in vivo studies [158,159,161]. Pretreatment of LPS-stimulated BV2 microglia cells with 30, 50, and 100 µM of urolithin B promoted anti-inflammatory effects by modulating pro-inflammatory markers, reducing nitric oxide, TNF-α, IL-6, ROS generation levels, and increasing HO-1 levels and gene expression [158]. These effects were also related to the suppression of NF-κB activity via inhibition of IκBα phosphorylation and AP-1 activity [158]. Urolithin B (0.7 mg/Kg, intraperitoneal route) in the ischemia/reperfusion damage model in Sprague Dawley rats reduced myocardial apoptosis and alleviated cardiac impairment through ROS reduction [161]. In H9c2 cells, urolithin B (5, 10, 20, and 40 µM) reduced ROS production through p62-Keap1 interaction and Nrf2 nuclear translocation with subsequent increase of HO-1, NQO1, and GSTP1 protein expression [161]. Moreover, urolithin B (10, 30, 50, and 100 µM) suppressed osteoclastogenesis through the reduction in ROS production in RANKL-stimulated osteoclast formation and activation in RAW264.7 cells with simultaneous attenuation of NF-κB, MAPK, and Akt signaling pathways and upregulation of Nrf2 and antioxidant enzymes [159].

Urolithins display several actions and a key role in the maintenance of intestinal health, acting by different mechanisms such as the regulation of intestinal microbiota, increasing beneficial bacteria such as *Lactobacillus*, *Akkermansia*, *Gordonibacter*, *Bifidobacterium* and *Clostridium*, and reducing bacterial infection with an improvement of intestinal barrier function [149]. In the DSS-induced intestinal inflammation in male Fischer rats, urolithin A at a dose of 15 mg/Kg/day for 25 days before damage induction ameliorated macroscopic and microscopic intestinal inflammation parameters as well as promoted an increase of bifidobacteria and lactobacilli and antioxidant properties as evidenced by ROS scavenging activity, downregulation of COX-1 and iNOS expression and NO production [162]. In a recent review was reported that several compounds, including urolithin A, can act on the aryl hydrocarbon receptor to protect and control intestinal inflammatory processes [163]. Urolithin A (40 µM) and urolithin B (5 µM) were also evaluated in human acute monocytic THP-1 cells and colon fibroblasts and promoted anti-inflammatory effects with a significant reduction of IL-1β, TNF-α, a downregulation of PGE_2_ and IL-8 and other regulators of cell migration and adhesion [164].

## 5. Conclusions

Considering the scientific evidence and proven relevance of coumarin derivatives as modulators of the Nrf2 signaling pathway to increase endogenous antioxidant defense and promote several pharmacological properties, several studies have been performed to design and synthesize new coumarin derivatives able to activate Nrf2 signaling and its antioxidant response, including oximes bearing coumarin derivatives [165], chalcone-coumarin hybrid compounds [166], coumarin-derived imino sulfonates [167], carbon monoxide release coumarin complexes [168], and coumarin-containing hybrids [9,10,11,12]. Coumarins have special physicochemical properties, such as π-π conjugation, low molecular weight, simple chemical structure, high bioavailability, and solubility, rich in electrons and charge properties, which could bind to many targeted proteins to produce a wide range of biological activities [6,7,168]. Moreover, coumarin derivatives have low toxicity and high versatility for synthetic transformation into a variety of functionalized derivatives, ensuring them a prominent role as lead compounds in drug design and development [6,7].

Several structure-activity relationship studies have been performed with coumarin derivatives in an attempt to identify the better chemical sites of coumarin structure to improve pharmacological activity. Recently, in an elegant study, 23 compounds based on osthole skeleton were designed, synthesized, and evaluated as potential Nrf2 agonists [169]. The study demonstrated that the introduction of an indole acetic acid structure in the C-7 position, after previous carboxylation, improved the Nrf2 agonistic activity of osthole, whereas the introduction of the tryptamine carbamate structure at the same position cannot improve the osthole activity as Nrf2 agonist [169]. On the other hand, the introduction of phenyl and methyl-phenyl groups had a significant and more pronounced effect enhancing agonistic properties of osthole, mainly when the substitution occurs at C-3 [169]. Similarly, a series of chemical modifications in multiple sites, mainly at the C-7 and C-8 positions of osthole were recently revised [170]. Structure-activity relationship evaluation demonstrated several C-7 and C-8 substitutions enhanced a lot of biological activities, mainly larvicidal, insecticidal, hemolytic, antitumoral, antimicrobial, antiparasitic, and neuroprotective properties [170]. Although it is not possible to conclude whether these chemical modifications are useful to improve Nrf2 activation properties, two compounds substituted at the C-7 position promoted better antioxidant and anti-inflammatory activity when compared with osthole [170].

Based on other coumarin derivative’s chemical structures, structure-activity relationship analysis demonstrated that chemical modification at C-6, C-7, and C-8 as well as the presence of an α,β-unsaturated carbonyl group or free hydroxyl at C-8 increases anti-inflammatory activity, including intestinal anti-inflammatory properties when compared with no substituted coumarins [84,168,171].

Among the coumarin derivatives here revised and displaying Nrf2/ARE signaling pathway activation to reduce oxidative stress and simultaneously promote intestinal anti-inflammatory activity, simple coumarins such as esculetin and its derivative 4-methylesculetin, daphnetin, and osthole, and the furanocoumarin imperatorin are attractive compounds for further backbone derivatization and screening as novel therapeutic agents potentially useful to control and prevent NCDs, particularly inflammatory bowel diseases. The efficacy at lower doses and the wide range of activities displayed by these highlighted natural coumarins together with their chemical versatility supporting several chemical modifications corroborate the use of the chemical skeleton of these coumarin derivatives for the drug design of bioactive compounds. The structure-activity relationship studies strongly suggest that C-6, C-7, and C-8 can be the main chemical sites of the coumarin derivatives structures for the design and synthesis of new compounds with Nrf2 activation and intestinal anti-inflammatory properties.

Although, other coumarin derivatives such as urolithin A, urolithin B, umbelliferone, esculin, fraxetin, scopoletin, and scoparone can be useful for further medicinal chemistry studies, additional in vitro and in vivo studies are necessary to better pharmacological characterization and evaluation of their potential as lead compounds.

Moreover, future clinical trial studies must consider health volunteers and ulcerative colitis and Crohn’s disease patients to determine the safety, efficacy, and impact of esculetin, 4-methylesculetin, daphnetin, osthole, and imperatorin in inflammatory bowel diseases. Even with these constraints, the studies here revised to highlight the potential of coumarin derivatives as lead compounds for the design, synthesis, and development of new compounds able to modulate the Nrf2 signaling pathway and promote intestinal anti-inflammatory activity, useful for the control, prevention, and to alleviate the symptoms of IBD and other NCDs.

## Figures and Tables

**Figure 1 pharmaceuticals-16-00511-f001:**
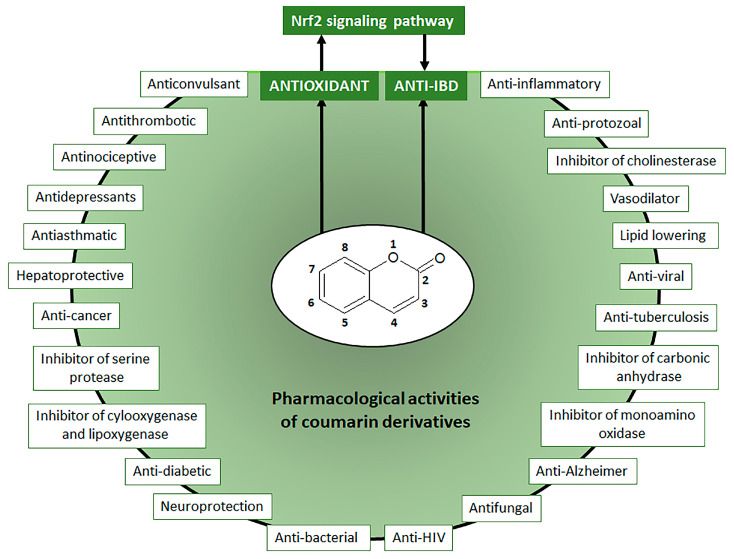
Main pharmacological activities of natural coumarin derivatives.

**Figure 2 pharmaceuticals-16-00511-f002:**
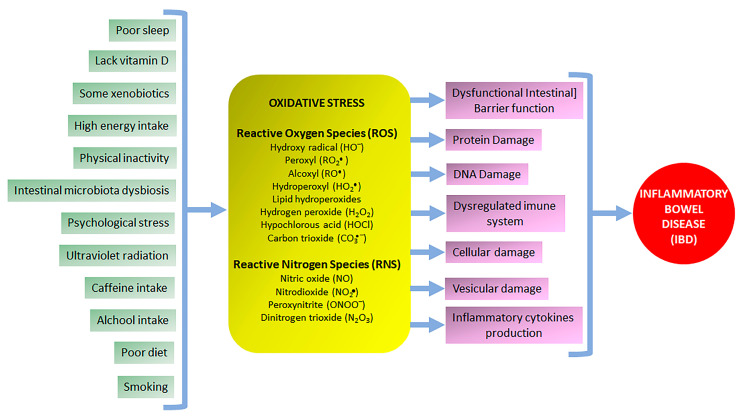
Extrinsic factors that trigger oxidative stress, cellular and molecular damage, and induce inflammatory bowel diseases.

**Figure 3 pharmaceuticals-16-00511-f003:**
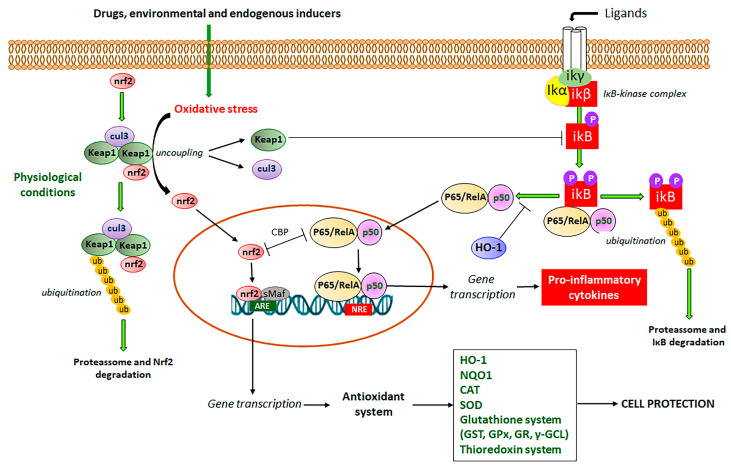
Nrf2 signaling pathway and its interaction with NF-κB transcriptional factor to control endogenous antioxidant system and inflammatory response.

**Figure 4 pharmaceuticals-16-00511-f004:**
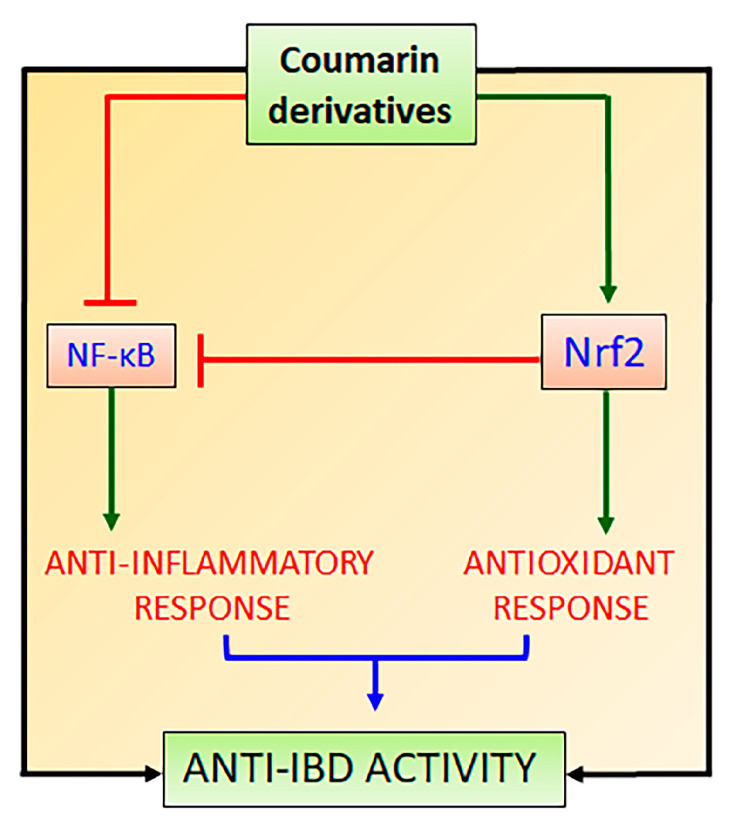
Natural coumarin derivatives action on Nrf2 and NF-κB signaling pathways to promote intestinal anti-inflammatory activity.

**Figure 5 pharmaceuticals-16-00511-f005:**
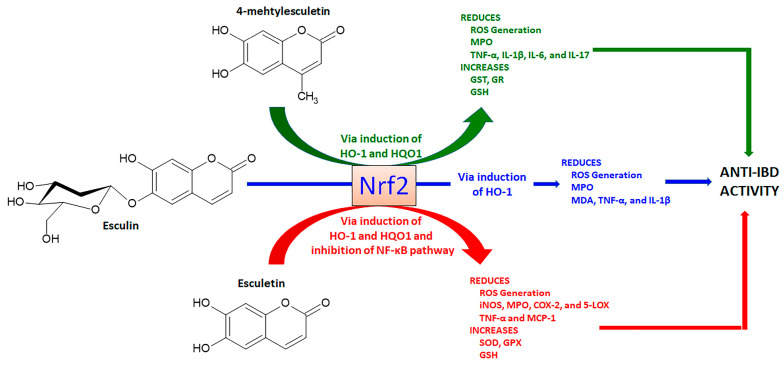
Chemical structures and main actions of esculetin, 4-methylesculetin, and esculin with intestinal anti-inflammatory activity by modulating the Nrf2 signaling pathway.

**Figure 6 pharmaceuticals-16-00511-f006:**
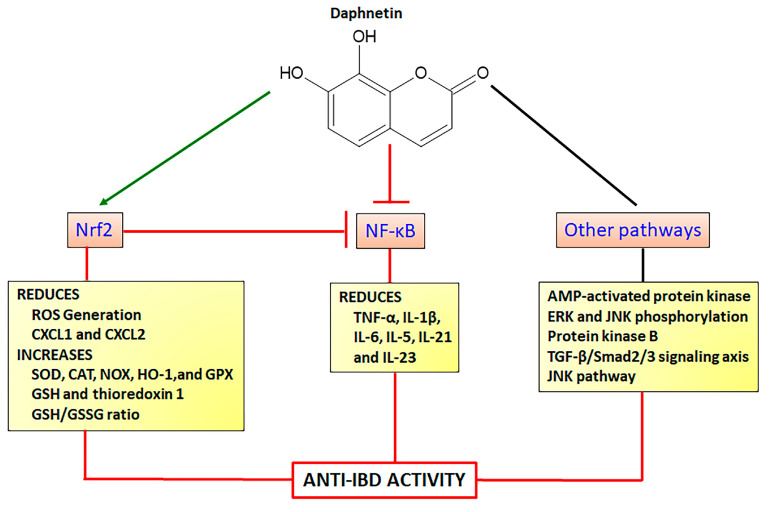
Chemical structure and main actions of daphnetin to promote intestinal anti-inflammatory activity.

**Figure 7 pharmaceuticals-16-00511-f007:**
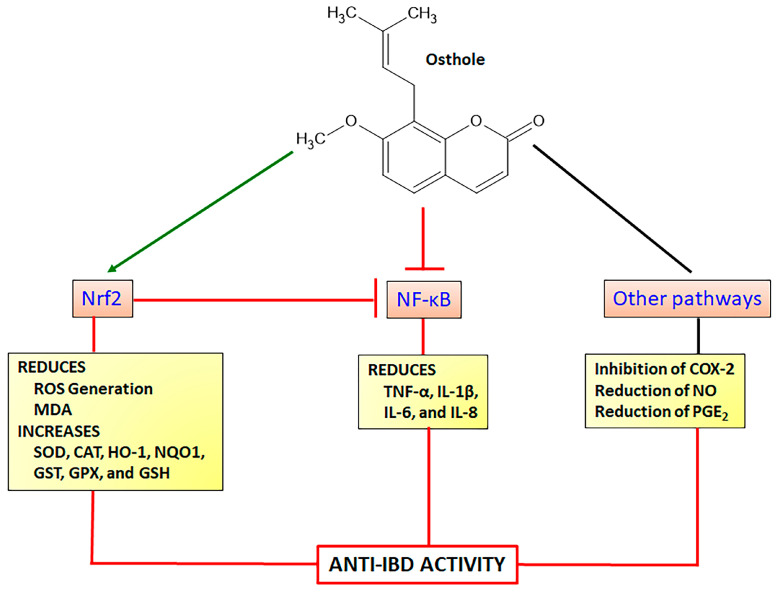
Chemical structure and main actions of osthole to promote intestinal anti-inflammatory activity.

**Figure 8 pharmaceuticals-16-00511-f008:**
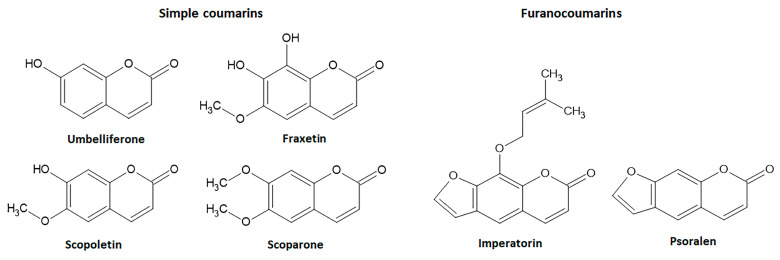
Chemical structures of simple and furanocoumarin derivatives with antioxidant and intestinal anti-inflammatory properties.

**Figure 9 pharmaceuticals-16-00511-f009:**
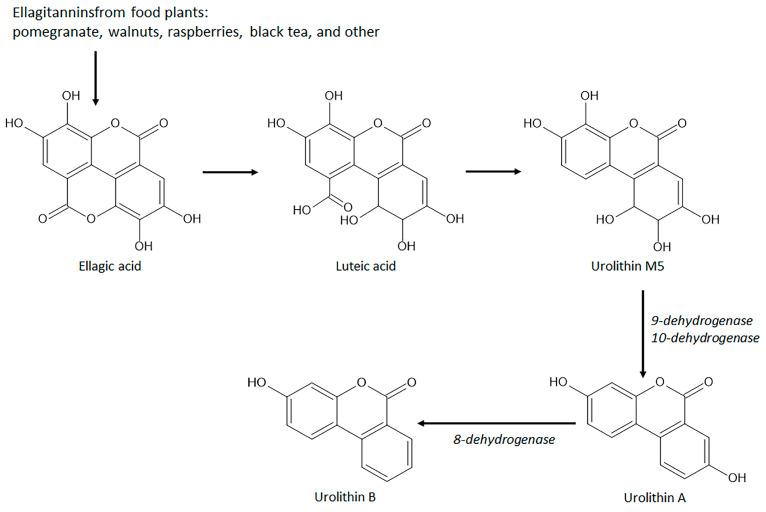
Chemical structures and biosynthetic route of gut microbiota-derived coumarins with antioxidant and intestinal anti-inflammatory properties.

## Data Availability

Data sharing not applicable.

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
