# Peer review of "Natural Coumarin Derivatives Activating Nrf2 Signaling Pathway as Lead Compounds for the Design and Synthesis of Intestinal Anti-Inflammatory Drugs"

_pharmaceuticals, 2023, doi:10.3390/ph16040511_

Round 1

Reviewer 1 Report

The article “Natural coumarin derivatives activating Nrf2 signaling pathway as lead compounds for the design and synthesis of intestinal anti-inflammatory drugs” discusses the importance of the transcription factor Nrf2 in maintaining the redox system and preventing non-communicable diseases, including Inflammatory Bowel Disease (IBD). Nrf2 activates the Keap1 signaling pathway, which inhibits NF-κB and promotes an anti-inflammatory response. Natural coumarins found in plants and food plants fermented by gut microbiota have been shown to activate the Nrf2/Keap signaling pathway and produce intestinal anti-inflammatory activity. The coumarin derivatives esculetin, 4-methylesculetin, daphnetin, osthole, and imperatorin are promising lead compounds for the design and synthesis of Nrf2 activators with intestinal anti-inflammatory activity, but further studies are needed to evaluate their potential efficacy and safety in IBD patients through clinical trials.

The exposed information is based on a sufficient number of references. In reference to the structure and length of the review, I think it is adequate and contains the right amount of information. The use of English is correct throughout the entire manuscript. The text is very clear and aligns perfectly with the scope of Pharmaceuticals.

Regarding specific changes, I propose:

- Line 664: The text in "(8-isopentenyloxypsoralen)" contains subtle highlighting. Please remove it.

- Line 793: Please change ROs to ROS.

Overall, I think the review is excellent and should be accepted for Pharmaceuticals publication after addressing the few changes proposed above.

Author Response

REVIEWER 1

Comments and Suggestions for Authors

Reviewer comment 1. The article “Natural coumarin derivatives activating Nrf2 signaling pathway as lead compounds for the design and synthesis of intestinal anti-inflammatory drugs” discusses the importance of the transcription factor Nrf2 in maintaining the redox system and preventing non-communicable diseases, including Inflammatory Bowel Disease (IBD). Nrf2 activates the Keap1 signaling pathway, which inhibits NF-κB and promotes an anti-inflammatory response. Natural coumarins found in plants and food plants fermented by gut microbiota have been shown to activate the Nrf2/Keap signaling pathway and produce intestinal anti-inflammatory activity. The coumarin derivatives esculetin, 4-methylesculetin, daphnetin, osthole, and imperatorin are promising lead compounds for the design and synthesis of Nrf2 activators with intestinal anti-inflammatory activity, but further studies are needed to evaluate their potential efficacy and safety in IBD patients through clinical trials.

Answer: I thank you very much for your comments, suggestions, and the time used to evaluate our review. Your comments were very kind and I am grateful for your consideration.

Reviewer comment 2. The exposed information is based on a sufficient number of references. In reference to the structure and length of the review, I think it is adequate and contains the right amount of information. The use of English is correct throughout the entire manuscript. The text is very clear and aligns perfectly with the scope of Pharmaceuticals.

Answer: I thank you very much for your comments.

Reviewer comment 3. Regarding specific changes, I propose:

- Line 664: The text in "(8-isopentenyloxypsoralen)" contains subtle highlighting. Please remove it.

                Answer: Thank you very much. In the new version of the manuscript this was corrected.

- Line 793: Please change ROs to ROS.

                Answer: Thank you very much. In the new version of the manuscript this was corrected.

Reviewer comment 4. Overall, I think the review is excellent and should be accepted for Pharmaceuticals publication after addressing the few changes proposed above.

Answer: I thank you very much again for your comments and suggestions. I inform you that the review was revised considering the comments of all reviewers, including a new English revision by a Native English.

                In the new version of the manuscript (revised) were included all modification suggestions by Reviewers 1 and 2 together with new changes based on additional English revision. All changes are highlighted in yellow. An additional discussion and bibliographic reference as suggested by reviewer 2 were included, which also are highlighted in yellow and found in lines 825 to 848 and 859 to 862. An additional reference (170) was also included in the new version of the manuscript.

Reviewer 2 Report

The review entitled "Natural coumarin derivatives activating Nrf2 signaling pathway as lead compounds for the design and synthesis of intestinal anti-inflammatory drugs" is a comprehensive review about coumarins activating Nrf2 pathway. Seems at the first glance an extension of the published review at https://doi.org/10.1155/2020/1675957 which diminish the enthusiasm for this review. There are some points which the author must work for example, in the title it says "lead compounds for the design and synthesis" what have been done in synthesis for each compound that the author report in the review? there are several articles published with that information that the author simply not report, for example for osthole there are a review which explain all the modification which have been done to increase their bioactivity published in: https://doi.org/10.1007/s00044-021-02775-w and the author do not mention, the author must expand the modification which have been done to increase their bioactivity for all coumarins. Also, the author must include a furanocoumarin which was not mentioned xanthotoxin see: https://doi.org/10.1002/ptr.7577. I would also like to see strong conclusions related to the title. 

The review is good but need major revision.

Author Response

REVIEWER 2

Comments and Suggestions for Authors

I thank you very much for your comments, suggestions, and the time used to evaluate our review. Your comments were very kind and I am grateful for your consideration.

Reviewer comment 1. The review entitled "Natural coumarin derivatives activating Nrf2 signaling pathway as lead compounds for the design and synthesis of intestinal anti-inflammatory drugs" is a comprehensive review about coumarins activating Nrf2 pathway.

                Answer: Thank you very much for this comment

Reviewer comment 2. Seems at the first glance an extension of the published review at https://doi.org/10.1155/2020/1675957 which diminish the enthusiasm for this review.

Answer: Yes, the review performed by Hassanein et al., 2020 (Coumarin as modulators of the keap1/Nrf2/ARE signaling pathway) was a very important scientific contribution to demonstrate the relevance of coumarin derivatives as Nrf2 activators. This review was included in our study and widely used to compose the discussion about coumarin derivatives acting on the Nrf2 signaling pathway. However, the review includes a different aim to include a discussion about coumarin derivatives targeting the Nrf2 signaling pathway to promote protective effects on non-communicable diseases (NCDs), particularly Inflammatory Bowel Disease (IBD). Since Nrf2 signaling pathway activation inhibits the NF-κB transcriptional factor with subsequent reduction of the production and release of the pro-inflammatory cytokines, the review focuses only on those coumarin derivatives targeting the Nrf2 signaling pathway and displaying intestinal anti-inflammatory activity, potentially useful for the development of new compounds to control and prevent Inflammatory Bowel Disease (IBD).

Reviewer comment 3. For example, in the title it says "lead compounds for the design and synthesis" what have been done in synthesis for each compound that the author report in the review?

Answer: Thank you very much for this comment. Some preliminary comments about chemical changes in coumarin derivatives structures were included in the final section of the review (lines 812 to 823 of the original manuscript), particularly those structural changes that improve both the anti-inflammatory and intestinal anti-inflammatory properties of coumarin derivatives. The review proposed to identify coumarin derivatives targeting Nrf2 as potentially useful as a scaffold for the development of the intestinal anti-inflammatory activity. Besides the discussion, there is a lack of scientific studies related to the synthesis of new drugs with intestinal anti-inflammatory activity based on coumarin derivatives. However, I included in the new version of the manuscript an additional discussion about this, considering the reviewer comments noted below. In the new version of the manuscript an additional discussion about the structure-activity relationship was included and can be analyzed in lines 825 to 848 and 859 to 862.

Reviewer comment 4. There are several articles published with that information that the author simply not report, for example for osthole there are a review which explain all the modification which have been done to increase their bioactivity published in: https://doi.org/10.1007/s00044-021-02775-w and the author do not mention, the author must expand the modification which have been done to increase their bioactivity for all coumarins.

Answer: Thank you very much for this comment and suggestion. In the new version of the manuscript, I included a general comment about this scientific contribution (See lines 825 to 848 and 859 to 862 in the new version of the manuscript), limiting those chemical changes on osthole molecule related to anti-inflammatory activity. The manuscript of Sun et al., 2021 (Osthole: an overview of its source, biological activity, and modification development) is an elegant review reporting several chemical modifications of the lactone ring and modification of the C7-substituted osthole derivatives. Those selective chemical changes related to osthole structure improving anti-inflammatory activity were included. On the other hand, all chemical modifications on the osthole chemical structure related by Sun et al., 2021 were related to other biological activities with no relation to an inflammatory process. This way, it is not possible a detailed discussion about the chemical modifications on the osthole structure improving, for example, larvicidal, insecticidal, hemolytic, cytotoxicity, antimicrobial, antitumoral, antiparasitic, and neuroprotective properties. The use of these chemical modifications it will be very speculative to discuss their potential effects on intestinal anti-inflammatory activity and/or Nrf2 activation. At the same time, I thank you very much for the suggestion about Sun et al., 2021 publication, because this review, using the keywords reported in the review, did not identify this manuscript. Several comments about this were included in the new version of the manuscript.

Reviewer comment 5. Also, the author must include a furanocoumarin which was not mentioned xanthotoxin see: https://doi.org/10.1002/ptr.7577. I would also like to see strong conclusions related to the title. 

Answer. Thank you very much for this comment. Xanthotoxin is a furanocoumarin acting on the Nrf2 signaling pathway with a wide range of pharmacological activities. However, xanthotoxin was not evaluated in the intestinal inflammatory experimental models. This is the reason that xanthotoxin was not included in the review. Considering the main aim of our review focusing on coumarin derivatives acting on Nrf2 and displaying intestinal anti-inflammatory activity, this furanocoumarin was not included in our review. As commented, our review is to propose coumarin derivatives as a lead compound for the development of intestinal anti-inflammatory drugs via activation of the Nf2 signaling pathway

Reviewer comment 6. The review is good but need major revision.

Answer: I thank you very much again for your comments, suggestion, and time used to evaluate our review. I inform you that the review was revised considering the comments of all reviewers, including a new English revision by a Native English.

In the new version of the manuscript (revised) were included all modification suggestions by Reviewers 1 and 2 together with new changes based on additional English revision. All changes are highlighted in yellow. An additional discussion and bibliographic reference as suggested by reviewer 2 were included, which also are highlighted in yellow and found in lines 825 to 848 and 859 to 862. An additional reference (170) was also included in the new version of the manuscript.

Round 2

Reviewer 2 Report

The review is suitable for Pharmaceuticals